# A study on formalizing the knowledge of data curation activities across different fields

**Yasuyuki Minamiyama**[1], **Hideaki Takeda**[2], **Masaharu Hayashi**[1], **Makoto Asaoka**[1], **Kazutsuna Yamaji**[1]*

1 Research Center for Open Science and Data Platform, National Institute of Informatics, Chiyoda-City, Tokyo, Japan, 2 Principles of Informatics Research Division, National Institute of Informatics, Chiyoda-City, Tokyo, Japan

* yamaji@nii.ac.jp

## Abstract

In recent years, with the trend of open science, there have been many efforts to share research data on the internet. To promote research data sharing, data curation is essential to make the data interpretable and reusable. In research fields such as life sciences, earth sciences, and social sciences, tasks and procedures have been already developed to implement efficient data curation to meet the needs and customs of individual research fields. However, not only data sharing within research fields but also interdisciplinary data sharing is required to promote open science. For this purpose, knowledge of data curation across the research fields is surveyed, analyzed, and organized as an ontology in this paper. As the survey, existing vocabularies and procedures are collected and compared as well as interviews with the data curators in research institutes in different fields are conducted to clarify commonalities and differences in data curation across the research fields. It turned out that the granularity of tasks and procedures that constitute the building blocks of data curation is not formalized. Without a method to overcome this gap, it will be challenging to promote interdisciplinary reuse of research data. Based on the analysis above, the ontology for the data curation process is proposed to describe data curation processes in different fields universally. It is described by OWL and shown as valid and consistent from the logical viewpoint. The ontology successfully represents data curation activities as the processes in the different fields acquired by the interviews. It is also helpful to identify the functions of the systems to support the data curation process. This study contributes to building a knowledge framework for an interdisciplinary understanding of data curation activities in different fields.

## Introduction

In recent years, with the trend of open science, there have been many efforts to share research data on the internet [1]. The main purpose of researchers sharing research data is to improve research efficiency, to increase verifiability, and to generate new knowledge by reusing research data [2–4]. Research data reuse is an essential act for researchers to achieve open science [5].

Research data reuse occurs when the data provider processes the research data to make it interpretable and reusable [6], and the data reuser uses the processed research data. The set of

**Competing interests:** The authors have declared that no competing interests exist.

activities that make research data interpretable and reusable is called data curation [7]. The sequence of the data curation process includes various tasks such as cleaning, documenting, standardizing, formatting, and associating metadata with relevant research data and codes [8]. The high-quality metadata given by these tasks and mutual understanding of the tasks makes published research data interpretable.

The practice of data curation has been developed mainly in fields such as life sciences [9], earth sciences [10], and social sciences [11]. Through historical efforts, tasks and procedures have been developed in these fields to implement systematic data curation [12]. With the increasing reliability and interpretability of research data, the research style of reusing others' research data is becoming the norm [13].

For interdisciplinary reuse of research data, research data must be interpretable by researchers from different fields [14]. The problem here is the difference in data curation, which depends on the field. First, data cleaning and related tasks are often tacit knowledge and not documented in data curation records [15]. Even if they were recorded, the granularity of the recorded information varies widely among the fields [16]. Moreover, even if the granularity of recorded information is partially the same, identification is often difficult due to different representations of tasks and procedures [17]. Even in those leading fields, research data reuse is often closed within the field [18]; This variation in the data curation activities by field reduces the interpretability of research data activities in different fields. Without a method to overcome this gap, it will be challenging to promote interdisciplinary reuse of research data.

To interpret the tasks and procedures performed in different fields at the same granularity, it is necessary to manage the term used for tasks and procedures in an interdisciplinary method. Methodologies for clarifying and systematically expressing certain knowledge have been studied mainly in the knowledge engineering field. Among them, applied ontology has been established and widely supported for constructing a conceptual system of knowledge [19]. Applied ontology has a possibility for interdisciplinary understanding for structural knowledge sharing of the data curation tasks and procedures.

This study aims to build a knowledge framework for an interdisciplinary understanding of data curation activities in different fields. For this purpose, we investigate the practices of data curation conducted in each field to interpret the tasks and procedures in different fields. We analyze existing vocabularies, incorporating insights from subject experts in each field to understand the structure of data curation activities. As a result, we formalize this knowledge as an ontology for structural knowledge representation. This study will help to improve and facilitate interdisciplinary data curation annotation practices.

## Literature review

Data curation tasks and procedures are commonly described with a research data lifecycle model [1]. In a research data lifecycle model, the decisions involved in a set of data curation are divided into abstracted steps [20]. By performing data curation according to a lifecycle model, the data provider can perform each data curation task and procedure with accuracy and the data reuser can understand in detail the methodology and workflow used [12].

Two frameworks, knowledge creation and knowledge transfer, are presented as perspectives to better understand the data curation that takes place at each stage of the life cycle model [21]. Regardless of the theoretical framework, the actual model is a mixture of both. Table 1 shows an example of the fields and steps involved in a representative research data lifecycle [22–30].

The "Steps" row contains the steps defined by each organization, starting from the top. The steps defined by each field differ in terms of granularity. It is not easy to standardize decisions at each step throughout the life cycle of research data [17]. The tasks and procedures included

**Table 1. List of data curation activities by field.**

| Name of Institutions/ Communities | CLARIN-NL | Data Curation Network | DataONE | Digital Curation Centre | DPCVocab | EMBL Australia Bioinformatics Resource | ICPSR | UK Data Archive | U.S. Geological Survey |
|---|---|---|---|---|---|---|---|---|---|
| Fields | Humanities/ Linguistics | Multiple | Earth Sciences | Multiple | Earth sciences/ Life sciences | Life sciences | Social sciences | Social sciences | Earth Sciences |
| Steps | A: Identification and assessment B: Development of a curation plan C: Curation D: Validation E: Archiving | Ingest Appraise/ Accept Curate Access Preserve | Plan Collect Assure Describe Preserve Discover Integrate Analyze | Conceptualise Create or receive Appraise & select Dispose Ingest Preservation action Store Access, use & reuse Transform | Ingest Representation Provenance management Systems management Data storage Policies Preservation Public access provision | collecting integrating processing analyzing storing sharing publishing finding | Proposal development and data management plan Project start-up Data collection and file creation Data analysis Preparing data for sharing Depositing data | Transfer of data Assigning processing standard Data processing Documentation processing Metadata creation Additional user information Publishing data Delivering data Preserving data | Plan Acquire Process Analyze Preserve Publish/ Share |

This list is an example of the fields and steps involved in a representative research data lifecycle. The "Steps" row contains the steps defined by each organization, starting from the top.

in each field are more diverse than the steps themselves, and there is no comprehensive list of tasks and procedures performed in data curation across fields. In one of the few efforts to formalize definitions of tasks and procedures across fields, the Data Curation Network has drafted a glossary of terms to be used in a survey of cross-disciplinary data curation activities in the U.S. [27]. This glossary is based on the existing glossary provided by the Digital Curation Centre (DCC), Society of American Archivists (SAA), CASRAI, RDA Data Foundation and Terminology Group, Digital Preservation Coalition (DPC), RDC (Research Data Canada), ICPSR, and practices in U.S. university libraries. Such efforts can be evaluated as potentially helpful in capturing the data curation tasks and procedures at the level of activities and supporting knowledge sharing. However, there still some issues: There is no unified protocol for how definitions are described, nor is there a clear distinction between persons and softwares as performers. The lack of formalization of the circumstances under which tasks and procedures are performed makes it difficult to determine the software. Also, it leads to less accurate interpretation by third parties.

## Objectives and hypotheses

In this study, we assume that certain commonalities exist between the activities carried out in each field and aim to formalize the interdisciplinarity of the knowledge that describes the activities. First, we analyze the existing vocabulary and organize the descriptions according to a logical structure. Next, we conduct interviews with data curators from several fields to evaluate the validity of the vocabulary description from an interdisciplinary perspective. Finally, we formalize the data curation activities using ontology techniques based on these two results.

## Materials and methods

### Vocabulary analysis

In this section, we analyze the existing vocabulary and organize the descriptions according to a logical structure. To interpret data curation tasks and procedures in different fields, we need

an interdisciplinary framework that can be used as a yardstick. As observed in Literature Review section, the Data Curation Network defines 47 vocabularies for the most important data curation activities derived from multiple lexical analyses. These vocabularies have been used in various fields of investigation and are highly comprehensive; we have chosen to use the Data Curation Network vocabulary as our working framework for these reasons. We analyzed the vocabularies by using the IPO (Input—Process—Output) model to interpret the logical structure of data curation activities. Table 2 shows a list of the 47 vocabularies subjected to analysis and the control structure expressed at the definition level.

In this analysis, we classified the control structure of the vocabulary into two categories based on the pairs of input and output information extracted from each vocabulary. The first category is sequential processing, in which the output information of activity becomes the input information of a different activity (35 vocabularies), and the second is occasional processing, in which activities are conducted independently from the time series (12 vocabularies). This classification is consistent with existing models [28], so we judged it to be appropriate as a working framework. However, the following three points should be noted:

1. **Lack of vocabulary corresponding to the output information** Some of the "generation" activities corresponding to the output information are not defined. For example, several activities have "data files" as input information, such as "Chain of Custody" or "File Validation," but the vocabulary for activities that output data files is not defined.

2. **Lack of a vocabulary with different hierarchies** There are parallel and sequential processes that require multiple inputs for some output information. However, some activities that aggregate multiple input information do not exist. For example, activities that have data files as input information ("Arrangement and Description," "Conversion," "Data Cleaning," "Data Visualization," "Deidentification," "File Format Transformation," "File Renaming," and "Interoperability") are a series of activities that aggregate these activities to create an individual processed data file. However, "File Download" targets the processed data file that aggregates a series of these activities.

3. **No staffing/software information is included** Each vocabulary does not include staffing information, so it is difficult to know the roles required to perform these activities. Additionally, some vocabularies are assumed to be processed by repository software, which may have influence depending on the software implemented.

## Field survey

We conducted a field survey of several organizations that conduct data curation activities in Japan. The purpose of the survey was to evaluate to verify the validity of the working framework by reviewing the data curators in each field. The survey was also designed to determine the actual staffing status, which was not revealed in the vocabulary analysis.

**Selection of survey participant.** First, we conducted interviews with the data curators at each organization. Table 3 shows an overview of the surveyed repositories.

In selecting interviewees, we collected as many fields of practice as possible. On this basis, we limited our interviewees to those who can provide the following verification method: They must have provided some form of documentation and/or the data curator's review. We asked the survey institutions to cooperate in writing for the field survey. Each institution responded in writing and in the body of an email, and we surveyed only those agreed institutions. As a result, we conducted these interviews with people committing these repositories; four institutional repositories, i.e., Global Environmental Database (GED), Data and Sample Research

**Table 2. Results of input–process–output analysis of data curation activity vocabularies.**

| No | Activity | Input information | Process | Output information | Control structures |
|----|----------|-------------------|---------|--------------------|--------------------|
| 1 | Authentication | Data depositor identity information | Authenticate the identity of data depositors | Data depositor's identity authentication results | Sequential |
| 2 | Chain of custody | Data files | Generate data file provenance information | Data file provenance information | Sequential |
| 3 | Deposit agreement | Deposit agreement application information | Verify that deposited agreement file is fit for data repository's policies and conditions | Verification results of deposited agreement file | Sequential |
| 4 | Documentation | Information describing any necessary information to use and understand the data | Generate all information describing any necessary information to use and understand the data | Data document file | Sequential |
| 5 | File Validation | Data files | Generate and verify checksums for data files / Verify the data file format | Checksum verification result of the data files / File Format verification results | Sequential |
| 6 | Metadata | Information about a dataset that is structured for purposes of search and retrieval | Generate necessary information about a dataset that is structured for purposes of search and retrieval | Metadata file for purposes of search and retrieval | Sequential |
| 7 | Rights management | Data document file | Verify that retention and copyright rights inherent in data files are consistent with policies and conditions for access and reuse | Verification results on data file ownership and copyright | Sequential |
| 8 | Risk management | Data files/Data document file | Verify that external constraints contained in data files are consistent with policies and conditions | Verification results of external constraints contained in the data files | Sequential |
| 9 | Selection | Verification results of deposit agreement/file format/data file ownership and copyright/external constraints contained in the data files | Verify that the results of the various verifications conform to the collection policy of the repository | Results of acceptance/rejection decision | Sequential |
| 10 | Arrangement and description | Data files | Reorganize data files according to standards and policies set by the repository | Data files (re-organized) | Sequential |
| 11 | Code review | Computer code | Verify the computer code | Verification results of the computer code | Sequential |
| 12 | Contextualize | Data document file/Metadata file for purposes of search and retrieval | Generate link information related to data files | Link information related to data files | Sequential |
| 13 | Conversion (Analog) | Analog data | Convert information into machine-readable format | Data files (converted into machine-readable format) | Sequential |
| 14 | Curation log | Execution results of the data curation process and executor information | Record changes made to the data and executor information during the data curation process | Information that records the execution results of the data curation process and executor information | Sequential |
| 15 | Data cleaning | Data files | Detect and fix (or remove) defects and errors in data files | Data files (cleaned) | Sequential |
| 16 | Deidentification | Data files | Redact or remove personally identifiable or protected information (e.g., sensitive geographic locations) contained in data files | Data files (deidentified) | Sequential |

(*Continued*)

**Table 2.** (Continued)

| No | Activity | Input information | Process | Output information | Control structures |
|---|---|---|---|---|---|
| 17 | File format transformations | Data files | Transform files into open, nonproprietary file formats | Data files (transformatted) | Sequential |
| 18 | Transcoding | Data files | Encode audio/video files in ways that optimize reuse and long-term preservation actions | Data files (encoded) | Sequential |
| 19 | File inventory or manifest | Data files | Verify the number of data files, file types (extensions), and file sizes periodically | Verification results of data files | Sequential |
| 20 | File renaming | Data files | Rename data files | Data files (renamed) | Sequential |
| 21 | Indexing | Data document file/Metadata file for purposes of search and retrieval | Crosswalk to descriptive and administrative metadata compliant with a standard format for repository interoperability | Metadata files that conform to the repository's standard format | Sequential |
| 22 | Interoperability | Data files | Format the data using a disciplinary standard | Data files (formatted) | Sequential |
| 23 | Peer-review | Data files/Data document file/Computer code | Validation of data files/data document file/computer code according to discipline-specific criteria by peers | Validation results of data files/data document file/computer code by peers | Sequential |
| 24 | Persistent Identifier | Data files/Metadata files that conform to the repository's standard format | Generate persistent identifier for data files / Set up redirection when necessary | Persistent identifier for data files / Redirect URL for data files | Sequential |
| 25 | Quality assurance | Data files/Data document file/Computer code | Validate data files/data document file/computer code according to the standards set by the repository | Validation results of data files/data document file/computer code | Sequential |
| 26 | Restructure | Data files | Organize and/or reformat poorly structured data files | Data files (restructured) | Sequential |
| 27 | Software registry | Data document file/Metadata file for purposes of search and retrieval | Maintain copies of modern and obsolete versions of software (and any relevant code libraries) | Copies of modern and obsolete versions of software (and any relevant code libraries) | Occasional |
| 28 | Contact information | Data document file/Metadata file for purposes of search and retrieval | Generate contact information for the data depositor and/or contact person / Update contact information for the data depositor and/or contact person | Contact information for the data depositor and/or contact person / Latest contact information for the data depositor and/or con-tact person | Occasional |
| 29 | Data citation | Metadata files that conform to the repository's standard format | Display of a recommended bibliographic citation | Recommended bibliographic citation text | Sequential |
| 30 | Data visualization | Data files/Data document file | Generate visualized data | Visualized data | Sequential |
| 31 | Discovery Services | Information on applying for connection to the discovery services/Metadata files that conform to the repository's standard format | Connect external discovery services | Discovery Service connection results | Sequential |

(*Continued*)

**Table 2.** (Continued)

| No | Activity | Input information | Process | Output information | Control structures |
|---|---|---|---|---|---|
| 32 | File download | Identifying information of authorized third parties/Metadata files that conform to the repository's standard format | Generate access URLs to data files by authorized third parties | Access URLs to data files by authorized third parties | Sequential |
| 33 | Full-text indexing | Data files | Generate text inherent in data file in search-engine-optimized formats | Full text information of the data files | Sequential |
| 34 | Metadata brokerage | Information on harvesting requests for metadata search and discovery services/ Metadata files that conform to the repository's standard format | Set harvesting requests for metadata search and discovery services | Results of harvesting settings for metadata search and discovery services | Sequential |
| 35 | Restricted access | Access permission information/Access URLs to data files by authorized third parties | Set access permissions for data files based on access permission information | Access URLs to data files by authorized third parties restricted by access authority in-formation | Sequential |
| 36 | Embargo | Embargo period information/Access URLs to data files by authorized third parties | Set an appropriate embargo period | Access URLs to data files with the embargo period set | Sequential |
| 37 | Terms of use | Metadata files that conform to the repository's standard format | Display information about the requirements or conditions for use provided to the end user of the data files | Information on the requirements or conditions for use of data files | Sequential |
| 38 | Use analytics | Data files/Data document file/ Metadata files that conform to the repository's standard format | Generate information on the frequency of data views, requests, and downloads ———————— Generate reuse metrics information such as data citations and impact measures for the data over time | Various usage information about data files | Occasional |
| 39 | Cease data curation | Information on data file storage and disposal plans | Plan for any contingencies that will ultimately terminate access to the data | Data Storage and Disposal Policy | Occasional |
| 40 | Migration | Data files | Transform obsolete file formats to new formats | Data files (migrated) | Occasional |
| 41 | Emulation | Copies of current versions of software (and any relevant code libraries) | Store and/or provide software to use the data files available in legacy systems | Software for emulation | Occasional |
| 42 | Secure storage | Data files | Back up data files on a regular basis | Backup data files | Occasional |
| 43 | File audit | Data files | Verify the digital integrity of data files | Verification results of digital integrity of data files | Occasional |

*(Continued)*

**Table 2.** (Continued)

| No | Activity | Input information | Process | Output information | Control structures |
|----|----------|-------------------|---------|---------------------|--------------------|
| 44 | Repository certification | A set of information about repository certification | Verify the technical and administrative capabilities of the repository by a trusted third-party accreditation body | Trusted third-party review results for repositories | Occasional |
| 45 | Succession planning | Information about the repository's long-term management plan | Develop a succession plan for the repository | Succession plan for the repository | Occasional |
| 46 | Technology monitoring and Refresh | Technical information about repository | Validate the performance of the repository against the latest technical requirements | Verification results of technical information | Occasional |
| 47 | Versioning | Data files | Generate version information for data files | Version information for data files | Occasional |

This table shows a list of the 47 data curation activity vocabularies subjected to input–process–output analysis and the control structure expressed at the definition level defined by the Data Curation Network.

**Table 3.  List of surveyed repositories.**

| Organization name | Repository name | Name abbreviation | Repository type | Field | Repository Description |
|---|---|---|---|---|---|
| The Center for Global Environmental Research, Earth System Division, National Institute for Environmental Studies | Global Environmental Database | GED | Institutional | Global environmental issues | The Center for Global Environmental Research (CGER) at the National Institute for Environmental Studies (NIES) has created a Global Environmental Database (GED), which comprises data and research results collected and compiled from natural and social sciences. The GED serves as a fundamental database related to global environmental problems with an emphasis on global warming and climate change. |
| Center for Statistics and Information, Rikkyo University | Rikkyo University's social survey data archive | RUDA | Institutional | Social sciences | Rikkyo University Data Archive "RUDA" aims to collect, organize, and store social survey data which are valuable public assets, and they make the datasets widely available for research purposes such as academic secondary analysis and educational use in classes. |
| Japan Agency for Marine-Earth Science and Technology | Data and Sample Research System for Whole Cruise Information | DARWIN | Institutional | Marine-earth science | On the "Data and Sample Research System for Whole Cruise Information (DARWIN)" the Japan Agency for Marine-Earth Sciences (JAMSTEC) disseminates information for data, rock samples, and sediment core samples obtained by its research vessels and submersibles, and the agency links to related databases. |

*(Continued)*

**Table 3.** (Continued)

| Organization name | Repository name | Name abbreviation | Repository type | Field | Repository Description |
|---|---|---|---|---|---|
| Japan Science and Technology Agency National Bioscience Database Center | Life Science Database Archive | NBDC archive | Institutional | Life science | The Life Science Database Archive maintains and stores the datasets generated by life scientists in Japan in a long-term and stable state as national public goods. The Archive makes it easier for many people to search datasets by metadata (description of datasets) in a unified format and to access and download the datasets with clear terms of use (see here for detailed descriptions). |
| National Museum of Japanese History | Knowledgebase of Historical Resources in Institutes | khirin | Institutional | Japanese history | "khirin (https://khirin-ld.rekihaku.ac.jp)" is the information infrastructure system that has been developed by the National Museum of Japanese History. "khirin" is an attempt to provide access to historical materials held by universities and museums on their networks as well as to offer data in a stable and sustainable manner in collaboration with the Japan Search. |
| National Institute for Materials Science | Materials Data Repository | MDR | Institutional | Materials science | MDR: Materials Data Repository is a data repository that hosts materials research data and publications. Discover various data and publications using metadata tailored for materials. MDR is operated by the National Institute for Materials Science (NIMS), Japan. |

(*Continued*)

**Table 3.** (Continued)

| Organization name | Repository name | Name abbreviation | Repository type | Field | Repository Description |
|---|---|---|---|---|---|
| National Museum of Ethnology | Digital Picture Library for Area Studies | DiPLAS | Project | Ethnology | The purpose of this project is to support the representatives of Grant-in-Aid for Scientific Research projects conducting research in various regions of the world (including Japan), and to contribute to the research advancement by promoting the digitization and creating photographic materials database. |
| The Research Organization of Information and Systems, National Institute of Polar Research; Tohoku University; Nagoya University; Kyoto University; Kyushu University | Inter-university Upper atmosphere Global Observation NETwork | IUGONET | Project | Upper atmospheric physics | We have three action plan in the second term (FY2015-) as follows: To provide the infrastructure and opportunity of the upper atmospheric research for users, in particular, in emerging countries. To provide our products and know-how for other fields and to nurture human resources who can develop future database and utilize it. To promote the use of various data in a wide range of fields and support the advanced integration science. |

This table shows the surveyed repositories overview, including organization name, repository name and abbreviation, repository type, field, repository description.

System for Whole Cruise Information (DARWIN), Knowledgebase of Historical Resources in Institutes (khirin), and Materials Data Repository (MDR) and two project-based repositories, i.e., Digital Picture Library for Area Studies (DiPLAS) and Inter-university Upper atmosphere Global Observation NETwork (IUGONET) from August to November 2020. We conducted additional interviews with those committing two institutional repositories, i.e., the Rikkyo University Data Archive (RUDA) and the Life Science Database Archive (NBDC archive) in August 2021. Each repository adopts various data curation models based on the nature and characteristics of the research data in each field. By comparing the models through an abstracted process, it is possible to extract commonalities and differences in structure. Each interview survey took approximately 1.5 to 2 hours. We used a topic guide to share the specific phase of data curation activities with the interviewee. In the topic guide, we set nine questions

referring to the previous study categories [31]. The interview results were assigned to our working framework under the authors' responsibility and checked by each interviewer. The topic guide template used for the interviews is shown in S1 File.

**Evaluation of the working framework.**   Next, we tallied the number of activities supporting the working framework in eight repositories to evaluate the validity of working framework. Fig 1 shows the support rates for interpreting the working framework in eight repositories. The tabulation work was divided into the following two steps.

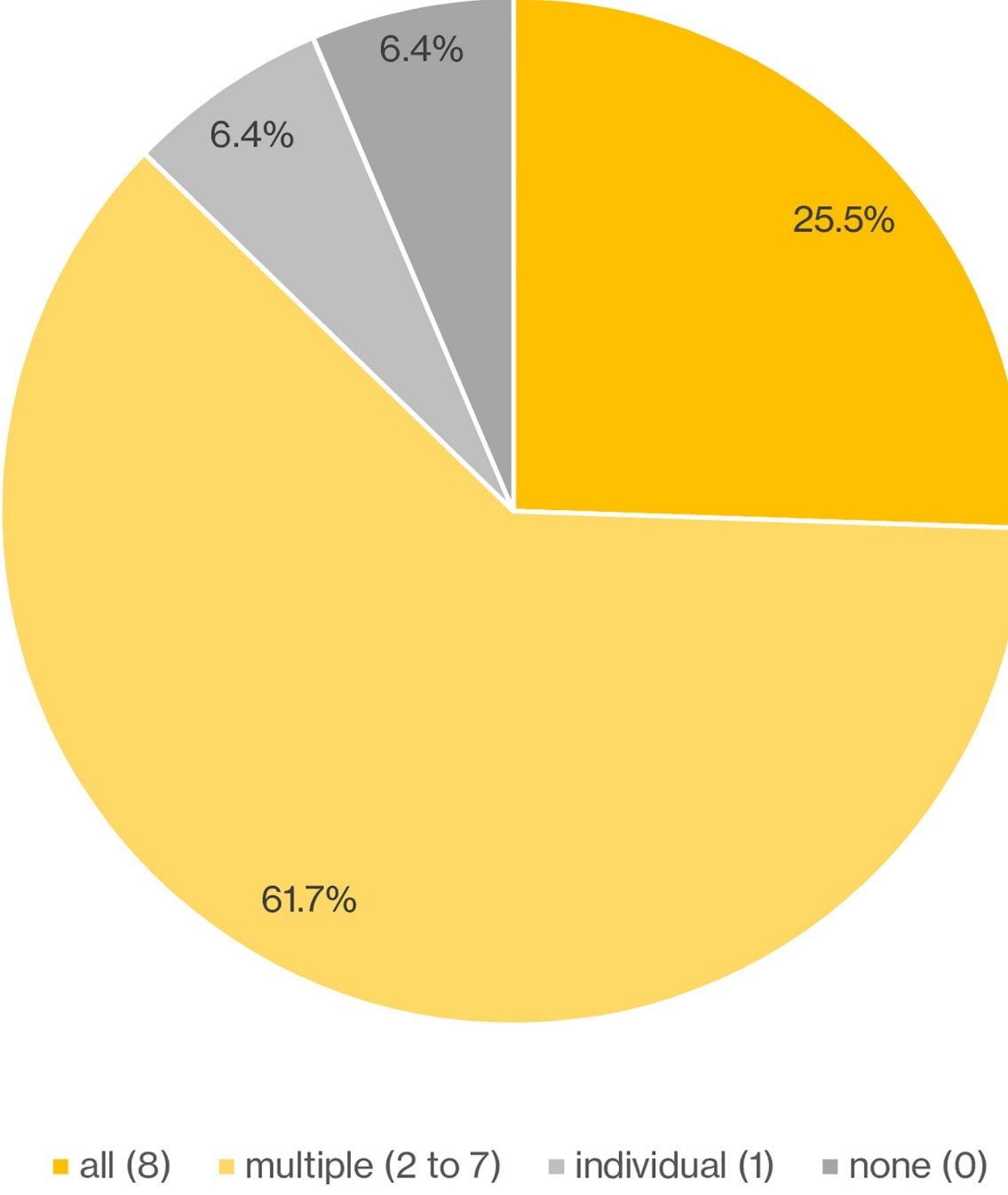

■ all (8)    ■ multiple (2 to 7)    ■ individual (1)    ■ none (0)

**Fig 1. Support rates for interpreting the working framework in eight repositories.** This pie chart shows the support rates for interpreting the working framework in eight repositories. For each of the 47 defined data curation activities, we classified the implementation number aggregated from each organization into four categories (all/multiple/individual/none).

*Step 1*: *Mapping of activities and working framework implemented in the eight repositories*. In step 1, we mapped the specific description of the activities and the data curators' information on the working framework for those activities for which we were able to identify a description of the rationale for the activities. Prior to mapping work, we read and referred to each organization's data curation process manuals and related documents for the rationale for the activities. For activities that were consistent with the interview results, we classified these activities as "Implemented". Although some of the activities were performed without manuals, we also classified these activities as "Implemented" with a "Survey participant" description in the "Rationale" columns. For activities with a description but only partially performed, we classified these activities as "Partially implemented". The activities classified as "Partially implemented" were mainly found when the vocabulary included multiple activities such as "generating and verifying checksums of data files" and "verifying file formats," as in "File Validation." For activities that could not be observed from the manual or from the interviews, we classified these activities as "Not implemented". The description of the rationale for all activities is shown in S1 Table.

*Step 2*: *Tallying the support rates of the working framework*. In step 2, we tallied the mapped activities as support rates of the working framework. We aggregated the implementation number of organizations by each activity. We also classified the implementation number by four categories (all / multiple / individual / none) from the perspective of interpretability. We note that we counted "b. Partially implemented" as one organization.

As a result, we found that approximately 87.2% of the activities in the working framework are supported across multiple fields. Among them, approximately a quarter of the activities were found to be fully supported across all fields.

**Observation of the variety of staffing status.** Additionally, we observed the variation in staffing. Table 4 shows an overview related to the staffing of each repository.

The roles defined by each repository are different, and there is no noticeable trend in the number of appearances. Each repository's data curation activities are conducted in different ways. For example, there are three staffing patterns in the "Data Cleaning" activities: the data holders themselves, the data curator(s), and the 2 or 3 parties working together. Some of these activities are covered by support systems or tools. For an interdisciplinary understanding of process execution, human actions and tool processes need to be viewed as different contributions to the process execution in the same actor.

## Formalizing the structure of data curation activities

Through vocabulary analysis, we organized the logical structure of data curation activities by using the IPO model. Furthermore, we observed the interpretability by subject experts in each field and the diversity of staffing roles conducting the activities. The two analyses revealed components for a structured understanding of data curation activities: input-output objects, hierarchical relationships among activities, and staffing. Since these relationships are complicated, it is not easy to represent the structure in a simple tabular form. Some model is needed to adequately describe these relationships.

To represent the structure of data curation activities, we adopt applied ontology as a model representation. Ontology is one of the methods for constructing conceptual systems used in the knowledge engineering field. The applied ontology provides a framework for knowledge sharing by clearly defining concepts and describing the logical relationships between concepts. Developing an ontology makes it possible to manage processes in which people and information systems are mixed.

**Development process.** To develop an appropriate ontology, it is recommended to follow some ontology developing procedure. Developing an ontology is not an easy task since

**Table 4. List of roles and number of appearances in eight repositories.**

| Repository name (abbreviated) | Roles | Number of appearances |
| --- | --- | --- |
| khirin | Researcher | 4 |
| | Related committee | 2 |
| | Center for Integrated Studies of Cultural and Research Resources | 27 |
| | Photographer | 2 |
| | System administrator | 1 |
| | Department of Rekihaku museum | 6 |
| | Department of internal database | 10 |
| | External organization | 1 |
| DiPLAS | Researcher | 2 |
| | Technical staff | 10 |
| | System administrator | 15 |
| | Data provider | 1 |
| | Project staff | 8 |
| | Digitization support staff | 1 |
| | Operation support staff | 1 |
| | Graduate students | 1 |
| | Review board | 1 |
| Materials Data Repository | Researcher | 6 |
| | Data system group | 14 |
| | Data service team | 13 |
| | System administration division | 1 |
| DARWIN | Researcher | 9 |
| | Data Management group | 42 |
| | Technician | 9 |
| | Navigation planning department | 2 |
| GED | Data provider | 14 |
| | Data curator | 29 |
| | Technical support staff | 1 |
| | Web application developer | 1 |
| RUDA | RUDA manager | 33 |
| | Research assistant | 10 |
| | Researcher | 5 |
| | System administrator | 1 |
| | Related committee | 2 |
| IUGONET | IUGONET manager | 23 |
| | Researcher | 16 |
| NBDC archive | Contact information staff | 9 |
| | Researcher | 14 |
| | Data curator | 17 |
| | System operator | 6 |
| | Repository manager | 1 |

explicating and formalizing the conceptual system behind the target system requires a very complex abstract thinking and reasoning. To ease the task, several procedures to develop an ontology are proposed. For the ontology development procedure, we followed the seven steps proposed by Noy & McGuiness [32]. In the actual work, we made several iterations between

Step 4 and Step 6 to maintain consistency with the hierarchical relationship. This ontology has 1748 axioms and 1086 annotation assertions generated as of version 1.1 (latest version). The results were validated using Protege ver. 5.5 with ELK 0.4.3 and also using Protégé ver. 4.3 with HermiT 1.3.8, Pellet 2.2.0, and FACT++ 1.6.2. The ontology is available at the following URL (https://purl.archive.org/curation-ontology).

*Step 1*: *Determine the domain and the scope of the ontology*. In this step, we determine the domain and the scope of the ontology to design an ontology. The decisions to be made include those for the domain to be covered by the ontology, the intended use of the ontology, and the development and maintenance of this ontology.

In our ontology, we represent the structure of data curation activities. The domain to be covered by this ontology is that of data curation. Providing structured data curation activities in a machine-readable format can support knowledge-sharing process between humans and information systems in a scalable manner. It is desirable to maintain the ontology through the collaboration of the data curators in each field and the ontologists who deal with knowledge sharing in information systems.

*Step 2*: *Consider reusing existing ontologies*. In this step, we consider reusing existing ontologies. Table 5 shows a comparison of the existing related ontologies.

As clarified in the Materials and Methods section, data curation activities contain both 'actions' by humans and 'processes' by software. Additionally, the performers implementing the same activity vary from field to field. The PROV ontology [33] with the best data model fit among the ontologies with these requirements.

The PROV ontology endorsed by W3C provides a set of classes, properties, and restrictions that can be used to represent and exchange provenance information generated by different systems and different contexts. Basic structure of the PROV ontology, the information is represented by three classes and their relationships: Activity, Entity, and Agent. In the case of data curation activities, the data curation process can be represented as the "Activity" class, the input information and output information as the "Entity" class, and staffing as the "Agent" class.

We mainly used the relationships defined in the PROV ontology to describe the relationships among Activities, Entities, and Agents. To identify metadata and curation records independently, we used the foaf:primaryTopic properties from the Friend Of A Friend (FOAF) ontology (http://xmlns.com/foaf/spec/)) as a complement.

*Step 3*: *Enumerate important terms in the ontology*. In this step, we enumerate important terms in describing the structure of data curation activities. Based on the analysis in the Materials and Methods section, we have chosen to extract many important terms in ontology from

**Table 5. Comparison of existing related ontologies.**

| Name | Domain | Scope | Remark |
|---|---|---|---|
| Activity Streams 2.0 | Social Data | Intended to be used with vocabularies that detail the structure of activities and that define specific types of activities | Highly scalable |
| PROV Ontology | Provenance Information | To represent and interchange provenance information generated in different systems and under different contexts | Actions performed by humans and processes performed by machines can be treated in the same framework |
| Wf4Ever Research Object Model 1.0 (extended the OAI-ORE Ontology) | Scientific investigation | The description of workflow-centric Research Objects | Specialized in describing workflow |

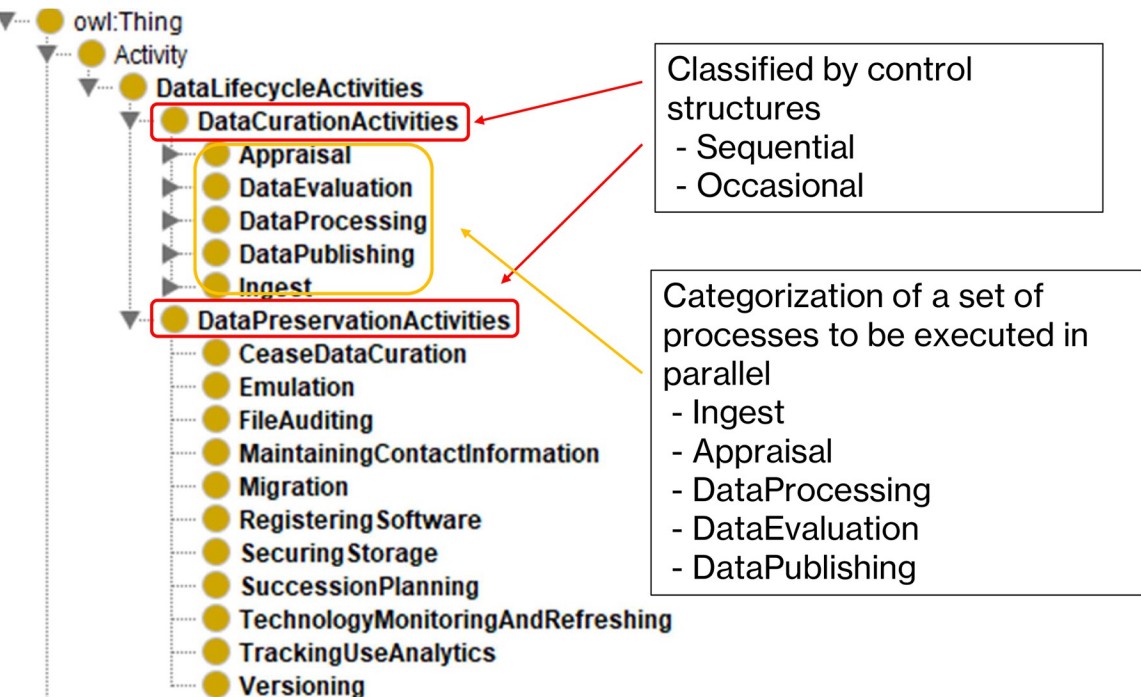

**Fig 2. Data curation process ontology structure.** This figure shows an overall structure of the data curation process ontology with a brief explanation.

the Data Curation Network vocabulary. We extracted many process descriptions, input information, and output information from the vocabulary to express the relationship between the structure of data curation activities with some modifications. We added four additional "Activity" vocabularies to organize the input-output information pairs: "SubmitData," "ActualDataProcessing," "MetadataProcessing," and "CreatingLandingPage" as the "Activity" class term. The criteria for the extraction are described in detail in Step 4.

*Step 4*: *Define the classes and the class hierarchy*. In this step, we define the classes and hierarchical relations of the ontology. Fig 2 shows the overall picture of this ontology's classes and hierarchical relations.

Before determining the logical hierarchical relationship between the classes, we performed a categorical division of the activities; as shown in the vocabulary analysis section, the extracted processes are a mixture of sequential and occasional processes. To separate the two types of activities with different control structures, we divided the classes into 'Data Curation Activities' for sequential processes and 'Data Preservation Activities' for occasional processes.

Next, we examined the logical structure of the 'Data Curation Activities'. Fig 3 shows the list of classes associated with each category.

We set the following five categories under 'Data Curation Activities': "Ingest," "Appraisal," DataProcessing," "DataE- valuation," and "DataPublishing." We already know that some sets of data curation activity are performed in parallel from the vocabulary analysis section. When managing this ontology, categorizing the process sets to be performed parallel helps interpretation. We set 23 processes under the five categories. In addition, two of the 22 processes have subclasses.

*Step 5*: *Define the properties of classes-slots*. In this step, we define the properties of the class-slots. Table 6 shows the list of properties used in this ontology.

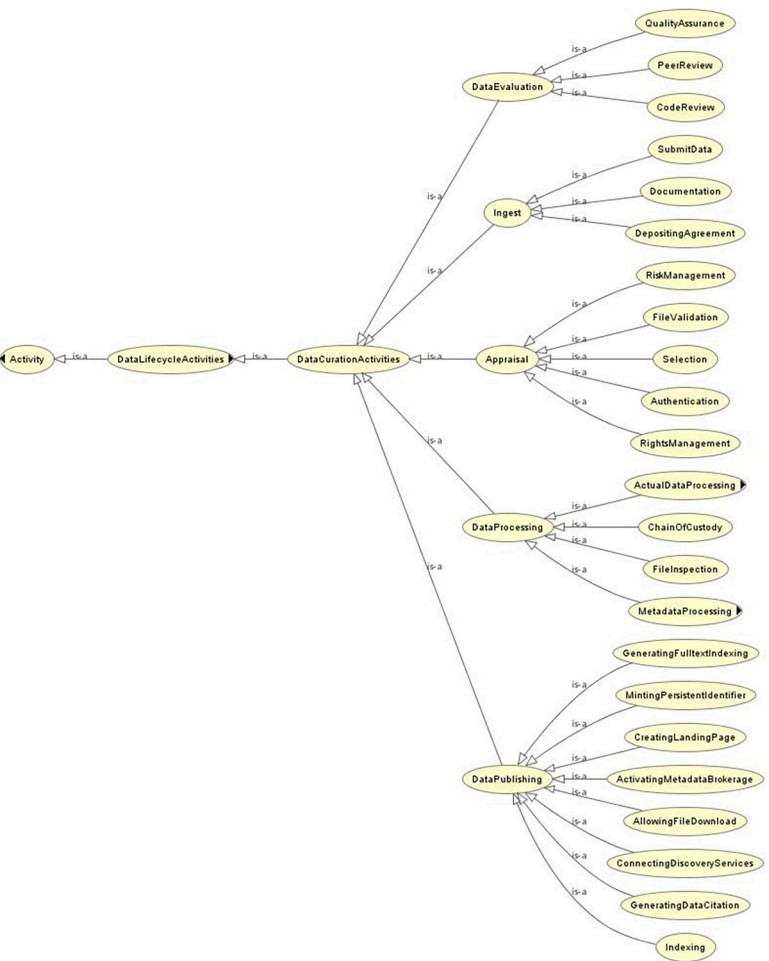

**Fig 3. List of classes by category for sequential data curation activities.** This figure shows the list of classes associated with each category for sequential data curation activity. We set the following five categories: "Ingest," "Appraisal," DataProcessing," "DataEvaluation," and "DataPublishing".

We adopted eight properties from the PROV ontology and one from the FOAF ontology. In describing the relationships in this ontology, we kept the description to the minimum necessary. In particular, the relationship between Activity and Entity is limited to "used" and "generated." In the reality of the structure of data curation activities, the relationship between Activity and Entity is far more diverse. For example, "CodeReview (Activity)" has the relationship of reviewing "sourceCode (Entity)."

However, having said that, describing the elaborate relationship intends to complicate the properties' semantics. Since the complexity of semantics may affect the structure of data curation activities in different fields, we adopted the above policy as the first step in this ontology.

**Table 6. List of properties used in data curation process ontology.**

| prefix | property |
|--------|----------|
| prov | used generated<br>wasAssociatedWith wasDerivedFrom wasInformedBy hadRole<br>Revision |
| foaf | primaryTopic |

*Step 6*: *Define the facets of the slots*. In this step, we define the value type, allowed values, number of values (cardinality), and other features of the values as the facets that can be set for each slot. Since facets' values can vary depending on the type of research data being included, it is necessary to accumulate data based on actual output information. Here, we have set tentative values for constraint types that align with the actual situation obtained from the field survey section.

*Step 7*: *Create instances*. In this step, we create an instance corresponding to the class of this ontology. Since this ontology abstracts the commonalities and differences in the structure of data curation activities, it does not address the description of instances, which are individual phenomena. The description of the actual structure of data curation activities is treated in the Results and discussion section.

## Results and discussion

This section shows how to use the data curation process ontology. Furthermore, this section also presents the specification of a data curation activities support function when using this ontology.

### Applications of the data curation process ontology

This section shows how to use data curation process ontology in three ways: "Representation of surveyed organizations," "Comparison of data curation activities across fields," and "New application for non-surveyed organization."

**Representation of surveyed organizations.** This section presents a representation using the ontology. Fig 4 shows the flow of data curation activities performed by RUDA, one of the institutions included in the field survey.

This flow diagram describes data dependencies for the data curation activities. The rows show the categories of "Ingest," "Appraisal," "Data Processing," "Data Evaluation," and "Data Publishing." The columns show five key entities: "Research Data," "Data Document," "Metadata," "Curation Record," and "Landing Page." Corresponding data curation activities and the generated entity are placed at the intersection of the rows and columns. The generated entity is connected to another data curation activities in which the entity is used by a "used" line. We note that this diagram describes agent information on the horizontal axis. Agents should be associated with each activity in the PROV ontology scheme. Since there are many agent-activity linkages, we describe agent information in the simplified form. The agent linked to the activity is described at the left-most column on the same row.

This diagram consists only of the classes defined in the data curation process ontology. Given any data curation activities that can be mapped to this ontology, we can represent any flow of data curation activities in a single model. The other examples for surveyed organization are available at the following URL (https://purl.archive.org/curation-ontology).

**Comparison of data curation activities across different fields.** This section compares data curation activities across different fields using the diagram expressed in the previous section. The possibility to describe activities in multiple fields in a single model contributes to comparing commonalities and differences across different fields. Fig 5 shows an example of the "Curation Record" comparison between IUGONET data curation activities (left) and RUDA (right).

The comparison shows that there are no "DepositAgreement" in the Ingest category and no "FileValidation" in the Appraisal category in the IUGONET data curation activities. The reason these activities have not been implemented in IUGONET is that IUGONET is a metadata distribution service that relies on the data provider for data access. There is no need to verify

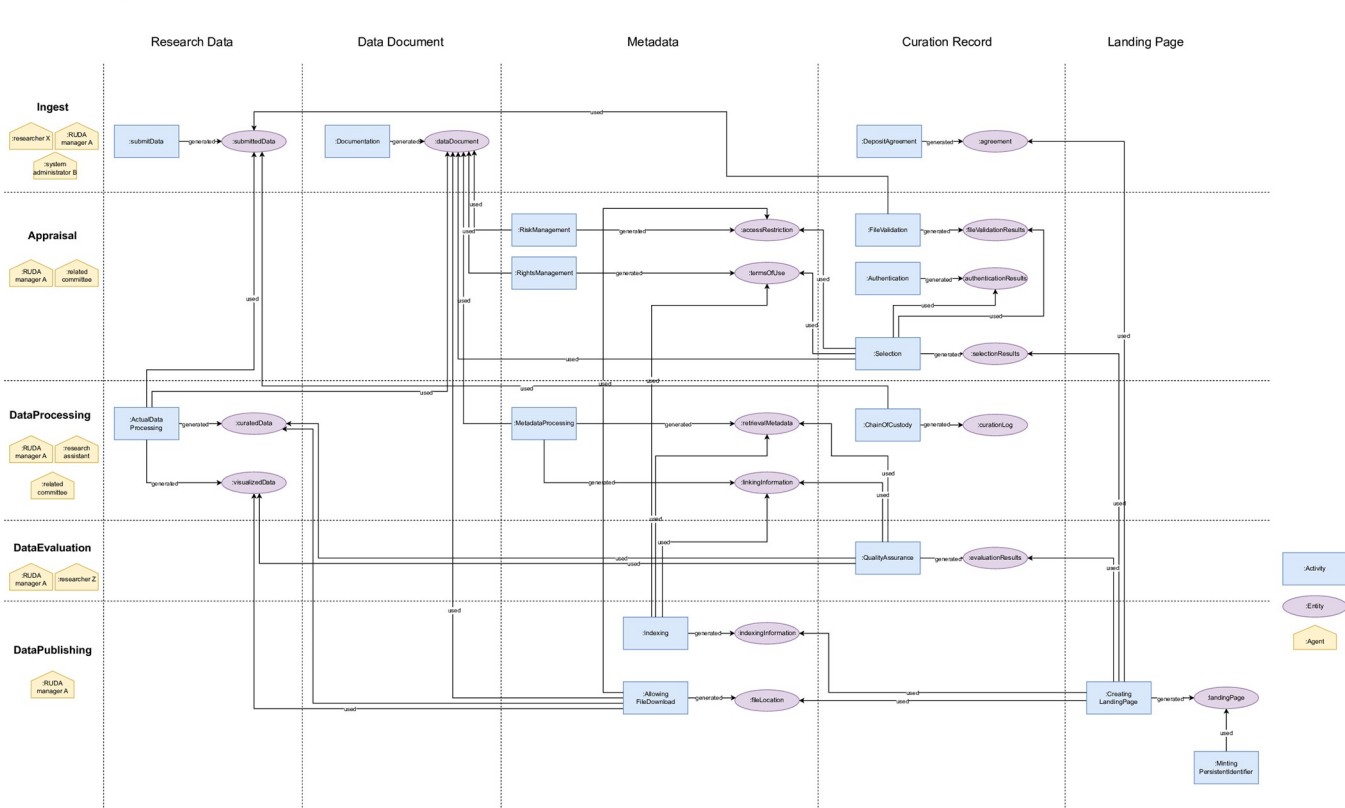

**Fig 4. The flow of RUDA's data curation activities.** This flow diagram describes data dependencies for the data curation activities. The rows show the categories of "Ingest," "Appraisal," "Data Processing," "Data Evaluation," and "Data Publishing." The columns show five key entities: "Research Data," "Data Document," "Metadata," "Curation Record," and "Landing Page." Corresponding data curation activities and the generated entity are placed at the intersection of the rows and columns. The generated entity is connected to another data curation activity in which the entity is used by a "used" line.

the data or to obtain permission for publication. Therefore, the "Authentication" is positioned as more important duty for the data curator in terms of comparison with other fields. Thus, identifying differences at the level of activities provides an opportunity to gain a deeper understanding of why the activity is or is not being implemented.

**New application for non-surveyed organization.** This section discusses the suitability of this ontology by applying this ontology for non-surveyed organization. To assess the general validity of this ontology, we attempted to annotate data curation manuals published by non-surveyed organization based on this ontology. As an annotation target, we chose GBIF (the Global Biodiversity Information Facility) (https://www.gbif.org/). The GBIF is an international network and data infrastructure funded by the world's governments and aimed at providing anyone, anywhere, open access to data about all types of life on Earth. The GBIF operates a portal site where participant nodes and their partners can apply for biodiversity data, and the JBIF (the Japan Initiative for Biodiversity Information) has been set up in Japan as a node organization. The GBIF provides details of the data curation activities to be carried out when registering on the portal on its web pages. The GBIF provides an overview of the procedure in "Quick guide to publishing data through GBIF.org (https://www.gbif.org/publishing-data)," with detailed procedures and guidance summarized mainly under the 'How-to' and 'Tools' tabs. In this assessment, we used to assess whether "Ingest," "Appraisal," "DataProcessing," "DataEvaluation," and "DataPublishing" in this ontology could comprehensively annotate the information on the page and the links contained on the page. We could not find a page

IUGONET
"Curation Record"

RUDA
"Curation Record"

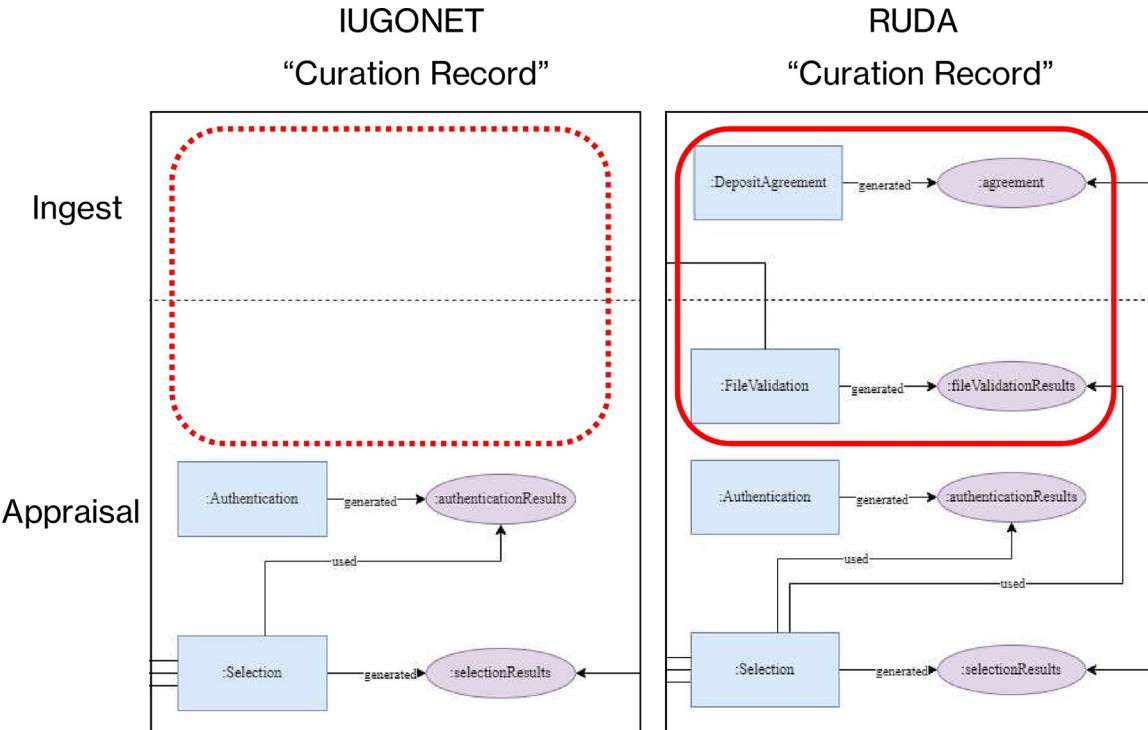

**Fig 5. Comparison of the data curation activities in different fields.** This figure compares the "Curation Record" of IUGONET data curation activities (left) with the data curation activities adopted by RUDA (right).

summarizing "DataPreservation" activities, so we searched the entire GBIF website for data preservation and management activities to assess these activities. This trial was conducted in March 2024. Table 7 shows the annotation results.

Table 7 shows the mapping of GBIF instances corresponding to each activity defined in the data curation process ontology. For comparison, the role information of the Agents and the value information of the three Entities (generated/researchObject/dataCurationResources) set in the ontology is described like "- as XXX." The trial results showed that all activities on the targeted pages were annotatable. We note that roles and values complemented by the authors to the manual context are marked with.

The "Ingest" category involves two Agents, data holders and data publishers. Registration with the GBIF requires an Agreement to be participated with an organization; the "DepositAgreement" activity is carried out in line with the Agreement agreed in advance by the data holders. Entities generated from the corresponding activities are "Resource metadata" and three types of data, as well as more detailed GBIF metadata and data papers. As explained before, the Agreement for deposit is included in the "Data publisher agreement" when registering data as an institution, so it does not appear during individual registration.

The "Appraisal" category involves two agents that continue to appear: data holders and data publishers. Data publishers carry out most activities, but "FileValidation" is carried out in advance by data holders to simplify the task on the publishers' side. Entities generated from the corresponding activities include authentication information, validation reports, terms of use, and access restrictions necessary for data registration decisions. Data holders generate validation reports; Data publishers are responsible for judging the results of the reports. Terms of use, and access restrictions align with the policy set by GBIF.

**Table 7. Application for GBIF data curation activities (as of March 2024).**

| Category | Data Curation Activities | GBIF Agent - as data curation process ontology role(s) | GBIF Entity (generated) - as data curation process ontology value(s) | GBIF Entity (used—researchObject) - as data curation process ontology value(s) | GBIF Entity (used—dataCurationResources) - as data curation process ontology value(s) | Related URL(s) |
|---|---|---|---|---|---|---|
| Ingest | SubmitData | Data holders | Resources metadata/Checklist data/Occurrence data/Sampling-event data | | | https://www.gbif.org/dataset-classes |
| | | - as dataDepositor | - as submittedData/sourceCode | | | |
| | Documentation | Data holders | GBIF metadata/Data papers | | GBIF metadata: https://gbif.jp/publishing/metadata/ Data papers: https://www.gbif.org/data-papers | |
| | | - as dataDepositor | - as dataDocument | | - as dataDocumentationPolicy | |
| | DepositAgreement | Data holders/Data Publishers | [Data publisher agreement] | | https://www.gbif.org/terms/data-publisher | |
| | | - as dataDepositor | - as depositAgreement | | - as selectionPolicy | |
| Appraisal | Authentication | Data Publishers | [Enquiry results via the form] | [Get an email address via the contact form] | | https://www.gbif.org/terms/data-publisher https://gbif.jp/en/publishing/support/ |
| | | - as administrator/dataCurator | - as authenticationResults | - as contactInformation | | |
| | FileValidation | Data holders/Data Publishers | Validation reports | Checklist data/Occurrence data/Sampling-event data | | https://www.gbif.org/tools/data-validator/about |
| | | - as administrator/dataCurator | - as fileValidationResults | - as submittedData/sourceCode | | |
| | RightsManagement | Data Publishers | CC0/CC BY/CC BY-NC | GBIF metadata/Data papers | | https://www.gbif.org/terms |
| | | - as administrator/dataCurator | - as termsOfUse | - as dataDocument | | |
| | RiskManagement | Data Publishers | Compliant with GBIF's "Memorandum of Understanding for the Global Biodiversity Information Facility: Paragraph 8 –Intellectual Property" | Compliant with GBIF's "Memorandum of Understanding for the Global Biodiversity Information Facility: Paragraph 8 – Intellectual Property" [Enquiry results via the form] GBIF metadata/Data papers Validation reports CC0/CC BY/CC BY-NC | | https://www.gbif.org/terms/data-publisher |
| | | - as administrator/dataCurator | [Enquiry results via the form] | - as dataDocument or submittedData | | |
| | Selection | Data Publishers | [Enquiry results via the form] | | "In principle, JBIF only accepts data owned by institutions and organisations." | https://gbif.jp/en/publishing/support/ |
| | | - as administrator/dataCurator | - as selectionResults | - as accessRestriction/authenticationResults/dataDocument/fileValidationResults/termsOfUse | - as selectionPolicy | |

*(Continued)*

**Table 7.** (Continued)

| Category | Data Curation Activities | GBIF Agent - as data curation process ontology role(s) | GBIF Entity (generated) - as data curation process ontology value(s) | GBIF Entity (used—researchObject) - as data curation process ontology value(s) | GBIF Entity (used—dataCurationResources) - as data curation process ontology value(s) | Related URL(s) |
|---|---|---|---|---|---|---|
| DataProcessing | ActualDataProcessing | Data holders | Validated data | Checklist data/Occurrence data/Sampling-event data | Darwin Core/EML: Ecological Metadata Language/BioCASe / ABCD | https://www.gbif.org/training |
| | | - as dataCurator | - as curatedData/visualizedData | - as submittedData/dataDocument | - as dataProcessingPolicy | |
| | - ArrangementAndDescription | X | X | x | x | |
| | - Conversion | – | – | – | – | |
| | - DataCleaning | X | x | x | x | |
| | - DataRestructuring | X | x | x | x | |
| | - DataVisualization | X | x | x | x | |
| | - Deidentification | X | x | x | x | |
| | - FileFormatTransformation | – | – | – | – | |
| | - FileRenaming | x | x | x | x | |
| | - Interoperability | x | x | x | x | |
| | ChainOfCustody | (Not specified) | (Not specified) | (Not specified) | | |
| | | - as dataCurator | - as submittedData | - as provenanceInformation | | |
| | FileInspection | (Not specified) | (Not specified) | | | |
| | | - as dataCurator | - as researchObject | | | – |
| | MetadataProcessing | | | | | |
| | • Contextualization | Data holders | GBIF metadata—Additional Information | (Not specified) | | https://gbif.jp/publishing/metadata/ |
| | | - as dataCurator | - as linkingInformation | - as researchObject | | |
| | • MetadataGeneration | Data holders | GBIF metadata | (Not specified) | | https://gbif.jp/publishing/metadata/ |
| | | - as dataCurator/dataDepositor | - as bibliographicInformation | - as researchObject | | |
| DataEvaluation | CodeReview | (Not specified) | | (Not specified) | | |
| | | - as dataCurator | | - as sourceCode | | |
| | PeerReview | [Peer reviewers] | | Data papers | | https://www.gbif.org/data-papers |
| | | - as peerReviewer | | - as curatedData/dataDocument/metadata/provenanceInformation/sourceCode | | |
| | QualityAssurance | [Data holders] | | Occurrence datasets/Checklists/Sampling-event datasets | | https://www.gbif.org/data-quality-requirements |
| | | - as dataCurator | | - as curatedData/dataDocument/metadata/sourceCode | | |

*(Continued)*

**Table 7.** (Continued)

| Category | Data Curation Activities | GBIF Agent - as data curation process ontology role(s) | GBIF Entity (generated) - as data curation process ontology value(s) | GBIF Entity (used—researchObject) - as data curation process ontology value(s) | GBIF Entity (used—dataCurationResources) - as data curation process ontology value(s) | Related URL(s) |
|---|---|---|---|---|---|---|
| DataPublishing | ActivatingMetadataBrokerage | GBIF API | | GBIF metadata | | https://techdocs.gbif.org/en/openapi/ |
| | | - as externalServiceProvider | - as indexingInformation/landingPage | | | |
| | AllowingFileDownload | [GBIF portal]/GBIF API (Registry API) | example: https://www.gbif.org/dataset/848586a4-a07b-4974-9f12-e1bbe0736a21 | GBIF metadata | | https://www.gbif.org/ https://techdocs.gbif.org/en/openapi/ |
| | | - as repositorySystem | - as fileLocation/versionInformation | - as accessRestriction/dataDocument/researchObject | | |
| | ConnectingDiscoveryServices | GBIF API | | GBIF metadata | | https://techdocs.gbif.org/en/openapi/ |
| | | - as externalServiceProvider | | - as indexingInformation/fullTextInformation | | |
| | CreatingLandingPage | Integrated Publishing Toolkit (IPT) | example: https://www.gbif.org/dataset/848586a4-a07b-4974-9f12-e1bbe0736a21 | GBIF metadata | | https://www.gbif.org/ipt |
| | | - as repositorySystem | - as landingPage | - as metadata/termsOfUse | | |
| | GeneratingDataCitation | [GBIF portal] | Example: Khidas K, Torgersen J (2020). Canadian Museum of Nature Bird Collection. Version 1.13. Canadian Museum of Nature. Occurrence dataset https://doi.org/10.15468/srfesr accessed via GBIF.org on 2020-09-23. | Author(s). Title. Version. Publisher. (Dataset type) (URL) via GBIF.org on (Date). | | https://www.gbif.org/faq?question=dataset-citation |
| | | - as repositorySystem | - as citationInformation | - as bibliographicInformation | | |
| | GeneratingFulltextIndexing | (Not specified) | (Not specified) | (Not specified) | | |
| | | - as [repositorySystem] | - as fullTextInformation | - as curatedData/dataDocument/submittedData | | |
| | Indexing | [GBIF portal] | [GBIF portal] | GBIF metadata | | https://www.gbif.org/search |
| | | - as [repositorySystem] | - as indexingInformation | - as bibliographicInformation/dataDocument/linkingInformation | | |
| | MintingPersistentIdentifier | Integrated Publishing Toolkit (IPT)/DataCite | Example: https://doi.org/10.15468/srfesr | GBIF metadata | | https://ipt.gbif.org/manual/en/ipt/latest/doi-workflow |
| | | - as repositorySystem | - as persistentIdentifier | - as indexingInformation/researchObject | | |

(Continued)

**Table 7.** (Continued)

| Category | Data Curation Activities | GBIF Agent - as data curation process ontology role(s) | GBIF Entity (generated) - as data curation process ontology value(s) | GBIF Entity (used—researchObject) - as data curation process ontology value(s) | GBIF Entity (used—dataCurationResources) - as data curation process ontology value(s) | Related URL(s) |
|---|---|---|---|---|---|---|
| DataPreservation | CeaseDataCuration | Data Publishers | Data publisher agreement | | | https://www.gbif.org/terms/data-publisher |
| | | - as dataManager | - as retentionPolicy | | | |
| | Emulation | (Not specified) | | | (Not specified) | |
| | | - as dataManager | | | - as softwareRegistry | |
| | FileAuditing | [GBIF] | | Validated data | | https://www.gbif.org/release-notes |
| | | - as dataManager | | - as curatedData | | |
| | MaintainingContactInformation | Data Publishers | GBIF metadata—Resource Contacts | GBIF metadata—Resource Contacts | | https://gbif.ip/publishing/metadata/#my-publishing-metadata-table-resource-contacts |
| | | - as dataManager | | - as contactInformation | | |
| | Migration | (Not specified) | | (Not specified) | | |
| | | - as dataManager | | - as curatedData | | |
| | RegisteringSoftware | [GBIF] | (Not specified) | (Not specified) | | https://www.gbif.org/resource/search?contentType=tool |
| | | - as dataCurationContributor | - as softwareRegistry | - as sourceCode | | |
| | SecuringStorage | [GBIF] | Automated monitoring—Downloads | Automated monitoring—Downloads | | https://www.gbif.org/system-health |
| | | - as dataManager | - as fileLocation or versionInformation | - as metadata/researchObject | | |
| | SuccessionPlanning | [GBIF] | GBIF Strategic Framework 2023–2027 | | | https://www.gbif.org/strategic-plan |
| | | - as dataCurationContributor | as successionPlan | | | |
| | TechnologyMonitoringAndRefreshing | [GBIF] | | "Technologies" | | https://github.com/gbif/portal16 |
| | | - as dataManager | | - as technicalInformation | | |
| | TrackingUseAnalytics | Collection managers | "Collection managers can trace usage and citations of digitized data published from their institutions and accessed through GBIF and similar infrastructures." | [GBIF portal] | | https://www.gbif.org/publishing-data |
| | | - as dataManager | - as usageResults | - as dataDocument/landingPage/researchObject | | |
| | Versioning | Integrated Publishing Toolkit (IPT)/GBIF | Example: Version 1.13 | GBIF metadata/Validated data | | https://ipt.gbif.org/manual/en/ipt/latest/versioning |
| | | - as dataCurationContributor | - as versionInformation | - as versionInformation/curatedData/submittedData | | |

The "DataProcessing" category involves only data holders who appear as Agents. Detailed manuals and various tools have been developed for each data curation activity to generate "Validated data" suitable for publication in the GBIF. "Contextualization" and "MetadataGeneration" activities are understood as part of the GBIF Metadata creation; Therefore, these activities are included in the Documentation carried out in the "Ingest" category. The data covered by the GBIF are not actual digitized data, so the "Conversion" activity is not executed. The activities corresponding to "FileFormatTransformation," "ChainOfCustody," and "FileInspection" could not be found in the manual; The reason may be that there is little or no need to handle these activities on the part of the GBIF side, as data holders carry them out independently.

The "DataEvaluation" category involves two Agents, data holders and peer reviewers. The "QualityAssurance" activity is dedicated to each data type (Checklist, Occurrence, and Sampling-event data). Data holders are required to be familiar with these manuals corresponding to the data they register and to produce high-quality data. The "PeerReview" activity is performed if a data paper has been created; GBIF provides several tools for creating data papers from the GBIF Metadata Profile, and some tools appear to support direct submission to data journals. The activity corresponding to "Code review" could not be specified in the manual.

The "DataPublishing" category involves three Agents: The Integrated Publishing Toolkit (IPT), the GBIF Portal, and the GBIF API. All Agents are categorized as SoftwareAgent, and each Agent corresponds to data registration, publication, and utilization. Except for "Activate-MetadataBrokerage" and "FullTextIndexing" activities, the Entity generated from each activity is understood as landing page elements within the GBIF. These Entities follow a prescribed format and are generated from the GBIF metadata. The activity corresponding to "FullTextIndexing" is not specified in the manual.

The "DataPreservation" category involves three Agents: Data publishers, GBIF, and Integrated Publishing Toolkit (IPT). Entities corresponding to "CeaseDataCuration" and "SecuringStorage" activities are predefined, and these activities output the execution logs in a form that meets daily requirements. Similarly, entities corresponding to "Versioning" and "TrackingUseAnalytics" activities will output an instance when data/information updates occur, and the "SuccessionPlanning" activity will output an instance every given year. The "FileAuditing" and "TechnologyMonitoringAndRefresh" activities do not have a corresponding Entity, but there are corresponding descriptions in the GBIF manual on the pages "Validated data" and "Technologies" respectively; These can be understood as activities that affect the entire data curation process. The "Emulation" and "Migration" activities were not specified in the manual.

As discussed in this section, annotating data curation activities using this ontology works well even for non-surveyed organization. Given that the annotations work well even for non-surveyed organization, we conclude that the ontology is suitably generic. Also, based on the annotation, it is possible to perform the representation and comparison shown in the previous section. This ontology can be helpful for mutual understanding of data curation activities in different fields.

## Specification of ontology-based data curation activities support functions

This section presents the specification of a data curation activities support function when using this ontology. Table 8 shows the mapping to the functions possessed by the repository software WEKO3 (https://rcos.nii.ac.jp/en/service/weko3/), which is a data publishing platform for researchers to publish research data and related materials and widely used in Japan.

WEKO3 supports basic data registration routes such as "SubmitData" and "FileValidation" and supports a wide range of metadata registration, editing, and publishing functions such as

**Table 8. Functional mapping with WEKO3.**

| Category | Data Curation Activities | Function name (WEKO3) | Remarks |
|---|---|---|---|
| Ingest | SubmitData Documentation DepositAgreement | Item registration (No function) (No function) | |
| Appraisal | Authentication FileValidation | Log-in Item registration | Except for file format validation |
| | RightsManagement RiskManagement Selection | (No function) (No function) (Partly) Workflow | Except for selection criteria support |
| DataProcessing | ActualDataProcessing ChainOfCustody FileInspection MetadataProcessing | (for journal article) Cover page creation Workflow (No function) Item registration/Item linking | Required processes vary by field |
| DataEvaluation | CodeReview PeerReview QualityAssurance | (No function) (No function) Item approval | |
| DataPublishing | ActivatingMetadataBrokerage CreatingLandingPage GeneratingDataCitation GeneratingFulltextIndexing Indexing AllowingFileDownload ConnectingDiscoveryServices MintingPersistentIdentifier | OAI-PMH harvesting / ResourceSync LandingPage displaying Citation creation Full-text indexing Index creation Download URL creation OAI-PMH harvesting / ResourceSync / Google Scholar metadata / schema.org DOI registration / CNRI handle | |

"MetadataProcessing," "ChainOfCustody," "QualityAssurance," and "DataPublishing." Whereas WEKO3 does not support some shareable processes related to data itself in each field such as "Documentation," "RightsManagement," "RiskManagement," "Selection," and "ActualDataProcessing." We note that activities related to the "DataPreservation" category are not included in the mapping, as WEKO3 does not include long-term data preservation in its scope.

As seen in Table 8, this ontology allows for comparisons at the functional level that can support data curation activities. This serves as a basis for the implementation of integrated data curation activities in conjunction with the various software developed in different fields.

## Conclusion

As the first step to build a knowledge framework for an interdisciplinary understanding of data curation activities in different fields, we investigated the practices of data curation conducted in each field. We analyze existing vocabularies, incorporating insights from subject experts in each field to understand the structure of data curation activities. As a result, we found that approximately 87.2% of the activities in the working framework are supported across multiple fields. Also, we realized that there needs a suitable model to describe the logical structure such as the relationships among Input-Output objects, processes, and staffing to accurately represent the data curation activity's structure in different fields. Based on the vocabulary analysis and survey results, we formalize the data curation activities using ontology techniques. To verify the usefulness and validity of this ontology, we represented and

compared the several actual data curation activity's structures. It is also the important contribution of this study to compare the activity's structure of eight diverse repositories in a single model. Also, we annotated data curation manuals published by non-surveyed organization based on this ontology. Given that the annotations work well even for non-surveyed organization, we concluded that the ontology is suitably generic. Finally, we showed that the ontology allows for comparisons at the functional level that can support data curation activities. This serves as a basis for the implementation of integrated data curation activities in conjunction with the various software developed in different fields.

By referring to this ontology, data managers can understand data curation activities at a higher level of abstraction. By comparing data curation practices in multiple fields, they may gain deeper insights into the data curation they practice themselves. Furthermore, it may be possible to incorporate activities not practiced in one's field in a formalized form to improve activities and respond to new challenges. From a similar perspective, educators in research data management can refer to this ontology to describe data curation activities more abstractly. For data curation activities that are highly field-dependent, this may lead to complementary general explanations and promote systematic understanding. It may also make it possible to efficiently incorporate practices from other disciplines when developing teaching materials for individual activities.

By elaborating on this ontology, future research could promote a better understanding of data curation activities. For example, we may develop building models to assess the maturity of data curation activities, analyze relationships between processes in more depth, and develop a vocabulary to express appropriate relationships further. Also, from a software engineering perspective, an integrated workflow construction based on this ontology can be considered. Currently, information systems used in various fields have been developed based on various design concepts; there needs to be a clearer perspective on which parts of the data curation activities are covered. Using this ontology makes it clear which processes can be covered by a certain information system and which are not. Furthermore, the semantics passed on between processes are defined, which may prevent important information from being missing.

Thus this study helps the stakeholders of data curation to interpret their procedural aspects of the research data curation and re-organize them in a more interpretable across different fields. As a result, it contributes the promotion of reusing research data for Open Science.

## Supporting information

**S1 File. Topic guide of "Questions related to data curation activities".**
(DOCX)

**S1 Table. List of data curation process description rationale.**
(XLSX)

## Acknowledgments

We would like to thank the following people for their cooperation in the field survey: Tomoko Shirai, Yoko Fukuda (National Institute for Environmental Studies), Kazuyo Fukuda, Hajime Kawakami (Japan Agency for Marine-Earth Science and Technology), Kosuke Tanabe, Chie Onodera, Asahiko Matsuda, Isao Kuwajima (National Institute for Materials Science), Makoto Goto (National Museum of Japanese History), Taku Iida, Yuzo Marukawa (National Museum of Ethnology), Yoshimasa Tanaka, Shiori Uchino (National Institute of Polar Research), Kaori Takahashi, Akiko Iwama, Wataru Nakazawa (Rikkyo University), Shigeru Yatsuzuka (Japan Science and Technology Agency National Bioscience Database Center). We also thank

Nobukazu Yoshioka (Waseda University) for giving us helpful advice on selecting analytical models. Moreover, we received valuable suggestions and assistance from Koichi Ojiro (formerly National Institute of Informatics).

## Author Contributions

**Conceptualization:** Yasuyuki Minamiyama, Hideaki Takeda, Kazutsuna Yamaji.

**Data curation:** Yasuyuki Minamiyama.

**Formal analysis:** Yasuyuki Minamiyama.

**Investigation:** Yasuyuki Minamiyama.

**Methodology:** Yasuyuki Minamiyama, Hideaki Takeda.

**Project administration:** Kazutsuna Yamaji.

**Resources:** Masaharu Hayashi, Makoto Asaoka.

**Supervision:** Hideaki Takeda.

**Validation:** Masaharu Hayashi, Makoto Asaoka.

**Writing – original draft:** Yasuyuki Minamiyama.

**Writing – review & editing:** Hideaki Takeda.

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
