## [Decision Letter · Decision Letter 0]

24 Jan 2024

PONE-D-23-30927A study on formalizing the knowledge of data curation activities across different fieldsPLOS ONE

Dear Dr. Minamiyama,

Thank you for submitting your manuscript to PLOS ONE. After careful consideration, we feel that it has merit but does not fully meet PLOS ONE’s publication criteria as it currently stands. Therefore, we invite you to submit a revised version of the manuscript that addresses the points raised during the review process.

We look forward to receiving your revised manuscript.

Kind regards,

Anna Bernasconi, PhD

Academic Editor

PLOS ONE

Journal Requirements:

3. You indicated that ethical approval was not necessary for your study. We understand that the framework for ethical oversight requirements for studies of this type may differ depending on the setting and we would appreciate some further clarification regarding your research. Could you please provide further details on why your study is exempt from the need for approval and confirmation from your institutional review board or research ethics committee (e.g., in the form of a letter or email correspondence) that ethics review was not necessary for this study? Please include a copy of the correspondence as an "Other" file.

4. Please provide additional details regarding participant consent. In the ethics statement in the Methods and online submission information, please ensure that you have specified (1) whether consent was informed and (2) what type you obtained (for instance, written or verbal, and if verbal, how it was documented and witnessed). If your study included minors, state whether you obtained consent from parents or guardians. If the need for consent was waived by the ethics committee, please include this information.

5. Please note that PLOS ONE has specific guidelines on code sharing for submissions in which author-generated code underpins the findings in the manuscript. In these cases, all author-generated code must be made available without restrictions upon publication of the work. Please review our guidelines at https://journals.plos.org/plosone/s/materials-and-software-sharing#loc-sharing-code and ensure that your code is shared in a way that follows best practice and facilitates reproducibility and reuse.

6. We note that your Data Availability Statement is currently as follows: "All relevant data are within the manuscript and its Supporting Information files."

Additional Editor Comments:

The authors are invited to design an evaluation on the suitability of the proposed ontology to annotate data curation-related metadata. This is an important comment from Reviewer 1. Please also address all the other comments made by the two reviewers. We will be ready to assess a revised version of the manuscript.

Reviewers' comments:

Reviewer's Responses to Questions

**Comments to the Author**

1. Is the manuscript technically sound, and do the data support the conclusions?

Reviewer #1: Partly

Reviewer #2: Yes

2. Has the statistical analysis been performed appropriately and rigorously? 

Reviewer #1: N/A

Reviewer #2: N/A

3. Have the authors made all data underlying the findings in their manuscript fully available?

Reviewer #1: Yes

Reviewer #2: Yes

4. Is the manuscript presented in an intelligible fashion and written in standard English?

Reviewer #1: Yes

Reviewer #2: Yes

5. Review Comments to the Author

Reviewer #1: The paper reviews data curation practices across a variety of disciplines, and based on this proposes an ontology to annotate data curation-related metadata. The proposed ontology is aimed to be used in an interdisciplinary setting, to allow cross-domain-compatible curation annotation.

The paper is written and structured reasonably well, although some sentences are hard to parse and I am not 100% sure I understood all details correctly. Some words are used incorrectly (e.g. "compliment" where "complement" is sensible). It is highly recommended to seek assistance of a native speaker (or a LLM) to improve the writing in future revisions.

Regarding the objective of the paper, I am not quite sure what the authors intended. I fundamentally endorse the stated goal of improving cross-discipline data curation annotation practices. Towards this, the authors are working on a very important and relevant problem, and their efforts are very welcome. It is my reading that the proposed ontology is aimed directly at facilitating this. However, if this is the case, I would have expected some form of evaluation on its suitability. Several examples on how it could be used would go a long way towards illustrating its relevance.

Overall, I believe the paper is not ready for publication in its current state. I would encourage the authors to continue their work to revise the paper and resubmit it at a later date. I would suggest that the authors focus the manuscript on the proposed ontology, add substantial documentation and examples of its relevance, and work out and describe deficiencies in the state of the art more stringently. In my judgement, the revisions required exceed what is possible in a major revision.

Other issues:

- The authors translated (mapped) the processes at the interviewed institutions themselves, but these translations are not available with the submission (that I was able to find). Interviewees are limited to Japan. It is unclear how much of a limitation this poses, since e.g. WEKO3 is a common factor among many institutions.

- The technical contribution centers around translating the Data Curation Network vocabularies into an ontology, where the authors have added a structuring categorization based on control structure and process grouping. Are alternative groupings possible? It appears that the ontology is sensible and useful, but this is not discussed in detail at all.

- I was not able to understand how consideration of staffing at the interviewed institutions contribute to the stated objectives. While this can be informative, the authors imply that it is possible to derive staffing requirements, which I find an invalid conclusion without considering further factors such as institution/repository size, submission rates, workflow overhead, etc.

- The authors critique that the Data Curation Network does not have strict hierarchy, but fail to stringently argue why this is needed in the first place. It appears that the problem the authors view is that it is not possible to create a complete process graph using on DCN information, but this is not stated in a comprehensible manner.

Reviewer #2: The study provides a valuable intellectual contribution to understanding data curation activities across domains. The paper is clearly written and presented data collection and analyses well. My only suggestions if possible would be expanding results and discussion. Future research and implications for data mangers and educators in this area would be helpful. Figures may need to be broken up into manageable parts. It is hard in a review to see how these might be presented at publication time, but there is a lot because these are complicated workflows. Overall, very important and challenging work.

6. PLOS authors have the option to publish the peer review history of their article (what does this mean?). If published, this will include your full peer review and any attached files.

Reviewer #1: No

Reviewer #2: **Yes: **Bradley Wade Bishop

While revising your submission, please upload your figure files to the Preflight Analysis and Conversion Engine (PACE) digital diagnostic tool, https://pacev2.apexcovantage.com/. PACE helps ensure that figures meet PLOS requirements. To use PACE, you must first register as a user. Registration is free. Then, login and navigate to the UPLOAD tab, where you will find detailed instructions on how to use the tool. If you encounter any issues or have any questions when using PACE, please email PLOS at <

---

## [Author Response · Author response to Decision Letter 0]

12 Mar 2024

We addressed all review comments in 'Response to Reviewers.docx.' We thank our editor and reviewers for their valuable feedback.

---

## [Editor Report · Decision Letter 1]

22 Mar 2024

A study on formalizing the knowledge of data curation activities across different fields

PONE-D-23-30927R1

Dear Dr. Minamiyama,

We’re pleased to inform you that your manuscript has been judged scientifically suitable for publication and will be formally accepted for publication once it meets all outstanding technical requirements.

Kind regards,

Anna Bernasconi, PhD

Academic Editor

PLOS ONE

Additional Editor Comments (optional):

The effort made by the authors to show how the proposed ontology can support the annotation of data curation manuals published by non-surveyed organizations enriches the contributions. The work undertaken by the authors tackles complex processes and deserves publication and attention in the research community.
---

## [Editor Report · Acceptance letter]

8 Apr 2024

PONE-D-23-30927R1 

PLOS ONE

Dear Dr. Minamiyama, 

I'm pleased to inform you that your manuscript has been deemed suitable for publication in PLOS ONE. Congratulations! Your manuscript is now being handed over to our production team.

Kind regards, 

on behalf of

Dr. Anna Bernasconi 

Academic Editor

PLOS ONE